# The Role of Small Heat Shock Proteins in Protein Misfolding Associated Motoneuron Diseases

**DOI:** 10.3390/ijms231911759

**Published:** 2022-10-04

**Authors:** Barbara Tedesco, Veronica Ferrari, Marta Cozzi, Marta Chierichetti, Elena Casarotto, Paola Pramaggiore, Francesco Mina, Mariarita Galbiati, Paola Rusmini, Valeria Crippa, Riccardo Cristofani, Angelo Poletti

**Affiliations:** 1Dipartimento di Scienze Farmacologiche e Biomolecolari (DiSFeB), Università degli Studi di Milano, Via Balzaretti 9, 20133 Milan, Italy; 2Unit of Medical Genetics and Neurogenetics, Fondazione IRCCS Istituto Neurologico Carlo Besta, Via Celoria 11, 20133 Milan, Italy

**Keywords:** MNDs, HSPBs, proteostasis, protein quality control, chaperones

## Abstract

Motoneuron diseases (MNDs) are neurodegenerative conditions associated with death of upper and/or lower motoneurons (MNs). Proteostasis alteration is a pathogenic mechanism involved in many MNDs and is due to the excessive presence of misfolded and aggregated proteins. Protein misfolding may be the product of gene mutations, or due to defects in the translation process, or to stress agents; all these conditions may alter the native conformation of proteins making them prone to aggregate. Alternatively, mutations in members of the protein quality control (PQC) system may determine a loss of function of the proteostasis network. This causes an impairment in the capability to handle and remove aberrant or damaged proteins. The PQC system consists of the degradative pathways, which are the autophagy and the proteasome, and a network of chaperones and co-chaperones. Among these components, Heat Shock Protein 70 represents the main factor in substrate triage to folding, refolding, or degradation, and it is assisted in this task by a subclass of the chaperone network, the small heat shock protein (sHSPs/HSPBs) family. HSPBs take part in proteostasis by bridging misfolded and aggregated proteins to the HSP70 machinery and to the degradative pathways, facilitating refolding or clearance of the potentially toxic proteins. Because of its activity against proteostasis alteration, the chaperone system plays a relevant role in the protection against proteotoxicity in MNDs. Here, we discuss the role of HSPBs in MNDs and which HSPBs may represent a valid target for therapeutic purposes.

## 1. Introduction

Motoneuron diseases (MNDs) are a subclass of neurodegenerative diseases (NDs) clinically characterized by motor unit alterations associated with various degrees of muscle atrophy and/or spasticity. These alterations are due to the dysfunction and/or loss of motoneurons (MNs) that directly (lower bulbar and spinal cord MNs) or indirectly (upper cortical MNs) control the contraction and relaxation of skeletal muscles. For example, spinal muscular atrophy (SMA), spinobulbar muscular atrophy (SBMA or Kennedy’s disease) are associated with selective lower MN loss; primary lateral sclerosis and hereditary spastic paraplegias are associated with upper MN death; instead, in amyotrophic lateral sclerosis (ALS) both upper and lower MNs are affected along with other neuronal types (e.g., neurons of the frontal and temporal lobes of the brain) [1,2,3,4]. MNDs can also display Parkinsonian symptoms such as muscle rigidity, tremors, and bradykinesia, as in the case of progressive supranuclear palsy, multiple system atrophy, and corticobasal degeneration [5]. MN dysfunction and death may arise from toxic events taking place in the affected cells, and/or be mediated by other surrounding cell types through non-cell autonomous mechanisms, which contribute to the onset and/or progression of the disease [6,7]. Similarly, the skeletal muscle cells, which are a direct MN target, could be either a primary site of disease or a contributing factor able to trigger MND-associated detrimental events, altering MN functions and survival capabilities [8,9,10,11,12,13]. In addition to muscle atrophy or spasticity, several other symptoms may be contemporarily present and often help in the differential diagnosis between upper, lower, or mixed MNDs [1,2,3,4].

Despite this clinical variability and the absence of a common genetic cause, the molecular bases of MNDs rely on specific and shared pathomechanisms involving common intracellular pathways. Inherited MNDs are characterized by well-defined gene mutations. In such cases, it may be feasible to differentiate between MNDs with loss of function (LOF) mutations, in which the protein encoded by the mutated gene has lost one of its normal functions, and MNDs with gain of function (GOF) mutations, in which the encoded protein has acquired a novel and potentially neurotoxic activity. However, this classification only provides a very simple view of the complexity of MNDs, since mutations in genes may result in both LOF and GOF [14].

The identification of the relative contribution of LOF and GOF to the disease is crucial to design an appropriate therapeutic strategy for MNDs. For example, while most MNDs are fatal and no cure is available at present, some therapeutic approaches have been recently developed for a specific form of juvenile MND clearly associated with LOF. This is the case of SMA, due to the LOF of the survival motoneuron (SMN) protein. The approved gene therapies for SMA have been successfully based on antisense oligonucleotides or on viral delivery of the complementary DNA encoding SMN, which are both able to restore normal levels of the functional protein counteracting the disease onset and progression in SMA patients [15,16]. These innovative approaches are paving the way for the development of novel treatments based on similar strategies. Unfortunately, despite the dramatic increase in our knowledge of the pathogenic mechanisms causing MNs loss in MNDs achieved in the past three decades, the potential clinical applications of these novel tools remain limited. The major limitation for the design of gene or pharmacological therapies is related to the wide diversity of the neurotoxic mechanisms causing GOF associated with MNDs. In fact, among the several neurotoxic events identified so far as the potential cause of dysfunction and/or death of affected cells, the finding of proper targets to be genetically hit for therapeutic purposes is still elusive. This is particularly relevant for the adult-onset forms of MNDs, in which all the deleterious processes acting on MNs may take years to become evident with the first appearance of the clinical signs of the disease [1,2,3,4]. Many transcriptomic and metabolomic analyses have been performed in order to define novel candidate genes that act as positive modifiers of MND progression or that can be utilized as diagnostic tools or targets for novel therapeutic strategies [17]. Notably, several mutated genes in MNDs encode for transcription factors or RNA-binding proteins, suggesting that the expression of some target genes may be altered and relevant for disease onset and progression [18,19,20]. However, the complexity of the mechanisms involved in MNDs, including significant parameters such as the stage of the considered MND, the source and type of tissue samples, and cell models utilized, might have limited the potential translation of data emerging from these analyses into effective therapeutic approaches [21].

Therefore, efforts in finding therapies in the MNDs field are specifically focused on counteracting a specific altered pathway in MNDs. One of these therapeutic targets is to tackle proteotoxicity, since this is one of the most common neurotoxic events involved in adult-onset MNDs [22].

Proteotoxicity is due to either the presence of proteins that are unable to properly fold and cannot reach their functional native state, becoming misfolded and accumulating, or defects in the protein quality control (PQC) system in assuring protein folding and turnover. Because of their size and the energetic demand required for the maintenance of their very long axonal processes, MNs are extremely sensitive to alterations in homeostasis and particularly to misfolded protein toxicity, such that most MNDs are also referred to as “proteinopathies” [23,24]. Thus, a highly regulated and stress-responding system able to deal with proteostasis alterations is mandatory for the functionality of neuronal cells. To counteract the proteotoxicity caused by misfolded proteins, these cells express—at high levels—several factors that are often redundant in their functions, which constitute the chaperone branch of the PQC system [25,26]. Among these, small heat shock proteins (sHSPs or HSPBs) represent a group of proteins that are specifically activated to respond to several intracellular stresses and that largely contribute to determining the fate of the misfolded proteins in terms of refolding or degradation [22,27,28,29,30,31]. Some HSPBs play relevant activities on MND-associated misfolded proteins; thus, this review will illustrate the importance of some HSPB family members for MN functions and their protective role against proteotoxic stresses in MNDs. Among the ten HSPBs, particular attention will be given to HSPB8, which represents the most valuable candidate for potential therapeutic applications in MNDs at present.

## 2. Sources and Mechanisms of Protein Misfolding in MNDs

Protein inclusions are a hallmark of several diseases associated with MN degeneration. They are generated by the accumulation of proteins that have lost their native conformation, undergoing misfolding. Protein misfolding can be triggered by: (i) specific amino acid mutations in the protein sequence; (ii) defects during the translation process; (iii) stress conditions (e.g., oxidative stress, heat shock, etc.) altering the biochemical properties of the protein; and (iv) extracellular sources (e.g., misfolded proteins transported into extracellular vesicles may exert a seeding effect via a prion-like mechanism when taken up from acceptor cells) (Figure 1).

Examples of proteins that misfold through these mechanisms have been found in MNDs. For instance, proteins that are prone to misfold and that are encoded by genes mutated in MNDs include the androgen receptor containing an elongated polyglutamine (ARpolyQ) tract in its N-terminus, which causes spinobulbar muscular atrophy (SBMA) [32], as well as superoxide dismutase 1 (SOD1), TAR DNA binding protein 43 (TDP-43), sequestosome-1 (SQSTM1/p62), and optineurin, which are all causative of different familial forms of ALS [33,34]. Several alterations in MNDs associated with misfolded protein accumulation have been recently reviewed in [24,35]. Notably, few reports associated proteotoxicity of mutated proteins with hereditary spastic paraplegias [36]. Nevertheless, it must be underlined that some mutated genes causative of hereditary spastic paraplegias are involved in autophagy regulation, which is known to have a great impact on the intracellular clearance of misfolded proteins [30,37,38,39]. Thus, LOF mutations in other components of the PQC system might associate with an accumulation of defective protein products, which ultimately results in their deposit into intracellular aggregates.

Even when not mutated, different proteins have been found accumulated in affected MNs in most of the sporadic forms of MNDs, likely due to undesired post-translational modifications (PTMs). This is the case of the TDP-43 protein, which is present in inclusions in the affected neurons of most ALS patients where it is characterized by abnormal phosphorylation, cleavage, and other erroneous PTMs even in its wild-type form.

It must also be noted that some gene mutations causing familial MNDs are linked with low penetrance, since some obligatory carriers do not manifest the disease and escape from identification and family pedigree design (e.g., in some cases related to the expansion of the G_4_C_2_ repeat present in the *C9ORF72* gene) [40,41,42]. Several of these forms are classified as sporadic MNDs, and when the corresponding mutations are not fully penetrant, these genes may be either considered causative or disease modifiers for MNDs [43].

Apart from the mechanistic trigger, upon misfolding, proteins organize their β-sheet domains in cross-β structures that are very stable, insoluble, and resistant to proteolysis [44,45,46]. Because of the generation of the cross-β structures and of the exposure of hydrophobic regions normally buried into the core of tertiary structures of properly folded proteins, as well as by the presence of highly disordered domains, misfolded proteins tend to clump together [47,48,49]. The monomeric and oligomeric misfolded proteins can either trigger further misfolding and accumulation of other proteins or sequester essential proteins for neuronal functions, generating a variety of protein aggregates [44,45]. Most of these structures may represent toxic species for neurons. Initially, highly dynamic droplet structures, also known as liquid–liquid phase partitioned structures or “condensates” [50,51,52], can be formed by the association of monomers and oligomers of misfolded proteins [47,53,54]. These dynamic structures are thought to be initially protective as they temporarily isolate misfolded proteins from the other cytoplasmic or nuclear components and prevent their toxicity [48,55,56,57]. Nonetheless, if not rapidly degraded or disassembled, these structures may mature into more insoluble aggregates and inclusions [47,53] with a great variability in terms of structure and biochemical properties, with potential toxic properties which are still largely debated [58].

Indeed, it is not clear whether protein aggregates are the result of an attempt of the cell to isolate proteins that might interfere with intracellular homeostasis. However, it is likely that, if not removed, protein aggregates alter intracellular functions by: (i) damaging membranes and organelles; (ii) sequestering factors involved in proteostasis maintenance; and/or (iii) disrupting the intracellular architecture and trafficking (Figure 1). As mentioned in the introduction, to circumvent these problems, cells maintain protein homeostasis, or proteostasis, through a network of factors that constitute the PQC system. Protein folding and degradation are regulated by the intracellular chaperones, which, together with the degradative pathways (ubiquitin–proteasome system (UPS) and autophagy) are the core components of the PQC system [25,26]. Notably, members of the chaperone family localize in all cell compartments, including nuclei, mitochondria, ER, and lysosomes [59,60]. Chaperones intervene to assist the folding of proteins during translation, to refold proteins that have been damaged or denatured upon stress, or to promote misfolded protein clearance via the UPS or autophagy. These functions aim to prevent the risk of misfolded protein condensation and precipitation into insoluble amorphous aggregates [60].

## 3. The Family of the Small HSPs (HSPBs) and Their Role in Protein Misfolding in MNDs

The Heat Shock Protein 70 (HSP70) subfamily constitutes the core system of the folding/refolding processes (see [61,62] for review), named “HSP70 machinery” (Figure 2). The HSP70 machinery performs iterative cycles in which the client protein to be folded binds to and is released from HSP70, thanks to the energy supplied by ATP. HSP70s are highly conserved and implicated in different intracellular pathways in addition to folding, such as protein disaggregation and degradation. To be involved in a specific activity, HSP70s cooperate with specific partners that regulate their functions, which are (co)-chaperones and cofactors (DNAJs/HSP40s and Nucleotide Exchange Factors (NEFs)), such as the Bcl-2 Associated athanogene proteins (BAGs) [61,62]. The HSP70 machinery interplays with the other chaperone families, including HSPCs/HSP90s, chaperonins (HSPD, HSPE, CCTs), and HSPBs. In particular, the HSPB family consists of ten low molecular weight protein members activated in response to proteotoxic (and other) stresses. HSPBs largely contribute to the HSP70-driven management of misfolded client proteins in their refolding or degradation [22,27,28,29,31]. The HSPBs share a highly conserved α-crystallin domain [22,27,28,63] while the N- and C-termini of HSPBs are generally poorly conserved. HSPBs considerably differ in terms of their intrinsic activities in facilitating refolding, disaggregation, or degradation of their target proteins. In addition, these activities can be externally modulated by different factors, such as the expression levels and subcellular localization of both HSPBs and client proteins, as well as by the biochemical/biophysical properties of the substrates [27]. While HSPBs can exist as monomers, they typically associate in high molecular weight oligomers (homo- and hetero-dimers as well as homo- and hetero-oligomeric complexes) [64,65,66] in a dynamic equilibrium between association and dissociation aimed at exerting their action. The various HSPBs highly differ in their tissue/cell -specific distribution, both in basal conditions and in response to different stress types. Only some HSPBs are present in the brain, and few are expressed in neurons or glial cells. Notably, misfolded protein production, causing proteotoxic stress, enhances the expression of some HSPBs already present and, also that of HSPBs normally absent in the affected cells [27]. Among all HSPBs, few have been linked to MNDs, mainly because their physiological role is crucial to maintain normal neuronal and muscular homeostasis.

### 3.1. HSPB1

HSPB1 is constitutively and widely expressed in human tissues. HSPB1 protects against thermal, oxidative, and proteotoxic stresses and its activity seems to be modulated by phosphorylation as a dimer or upon oligomerization in high molecular weight complexes [67,68,69]. In response to stress, HSPB1 is involved in protein refolding or degradation either via the UPS or autophagy [70,71]. Upon proteasome blockage or heat shock, HSPB1 relocates at the perinuclear region, in aggresome structures, or in the nucleus, in granules, where it favours the refolding of heat-denatured proteins. Although both heat shock and proteasome inhibition are able to induce changes in the phosphorylation and oligomerization state of HSPB1, it seems that HSPB1 relocation occurs independently from this PTM. Notably, it has been shown that the co-chaperone BAG1 negatively regulates HSPB1 activity in nuclear proteostasis [72,73]. HSPB1 also acts as an anti-apoptotic agent at different levels, since it is capable of sequestering pro-apoptotic factors of the Bcl2 family and the cytochrome C released after mitochondrial membrane permeabilization, to prevent APAF-1 and the caspase cascade activation [74,75,76,77]. In addition, HSPB1 modulates the assembly and protects the integrity of the cytoskeleton, preventing potential damage to its structure [78,79,80]. This activity is particularly relevant in skeletal and cardiac muscles [81,82,83]. Mutations of the gene encoding HSPB1 have been associated with different neuromuscular diseases; in particular, autosomal dominant hereditary distal axonal neuropathies, including Charcot Marie Tooth disease type 2 (CMT2) and distal hereditary motor neuropathy (dHMN). Notably, signs of upper MN involvement can also be detected in patients carrying *HSPB1* gene mutations, resulting in clinical phenotypes typically found in MNDs such as ALS [84,85]. Importantly, several actions of HSPB1 have been described in the brain, where it counteracts the formation of misfolded protein species [86,87,88] that may become toxic to neurons, but particularly in other neural cells, especially in astrocytes. Indeed, HSPB1 has been shown to be upregulated in glial cells of the spinal cord of transgenic (tg) mouse models of ALS (tg SOD1 G93A mice), and its levels gradually increased during disease progression [89,90,91]. Notably, in a reconstituted astrocytic-MN co-culture model mimicking ALS, HSPB1 overexpression in astrocytes attenuated SOD1-G93A toxicity in co-cultured MNs [92]. Despite this evidence, the overexpression of HSPB1 in a tg SOD1-G93A mouse model failed to rescue both the altered motor phenotype and the aberrant biochemical properties of mutant SOD1. HSPB1 overexpression did not modify disease onset and progression and had no effects on animal survival [90]. Of note, mice overexpressing HSPB1 were found to be more resistant to spinal cord ischemia [90]. Therefore, it is likely that HSPB1 protective activity against misfolded proteins is not sufficient to counteract the massive chronic MN injury exerted by misfolded mutant SOD1 in ALS. HSPB1 is also involved in stress granule maintenance and liquid–liquid phase transition process, two interconnected processes that, when altered, have been implicated in MNDs. In particular, HSPB1 exerts a chaperoning function on Fused in Sarcoma, a protein that undergoes liquid–liquid phase separation and, when mutated, associates with ALS. Similarly, HSPB1 regulates liquid–liquid phase separation and aggregation of TDP-43 and HSPB1 levels have been found decreased in neurons displaying TDP-43 pathology [93,94]. In addition, a brief report showed increased HSPB1 expression in a differential expression analysis on transcripts obtained from skeletal muscle of three different mouse models of SBMA [95]. HSPB1 protective role has been better investigated in other NDs: for instance, HSPB1 enhanced expression by indole compounds associated with beneficial effects on mouse models of Alzheimer disease and polyQ-associated spinocerebellar ataxia [96,97].

### 3.2. HSPB2 and HSPB3

Differently from other HSPBs, HSPB2 is not induced by heat shock, even if this type of stress affects its solubility [98,99]. However, HSPB2 still retains the ability to counteract some deleterious effects of heat stress [100]. Of note, the response to stresses includes modification of the intracellular distribution of HSPB2 [101,102], possibly as a mechanism to mediate its protective action in stressed cells. Remarkably, in the nucleus HSPB2 can generate droplets via liquid–liquid phase separation [103] and can specifically interact with HSPB3 [104,105]. No disease-associated gene mutations have been reported so far for HSPB2, and few data suggest a role of this HSPB in neurological disorders [106,107]. A recent study proposed a correlation between HSPB2 (together with AK4 and IGFBP5) and a faster decline of cognitive function in the older population, associating this with microstructural changes in the brain (i.e., alterations in frontal and temporal white matter regions) [108]. So far, only one report has demonstrated a potential involvement of HSPB2 in MNDs, showing an increased expression level of HSPB2 in SBMA. This upregulation of *HSPB2* gene expression was confined to the skeletal muscle (quadriceps) of affected male mice at post-symptomatic advanced disease stages, while no significant changes were observed in the spinal cord [11]. However, it must be noted that the number of MNs in the spinal cord is dramatically lower compared to the other cell types present in this region; therefore, specific changes occurring in these neuronal types could be hard to identify by differential gene expression analysis because the HSPB2 transcripts may be diluted in the samples by other transcripts coming from unresponsive neuronal and glial cells. Unfortunately, no immunolocalization data that would have provided evidence for a potential involvement of HSPB2 in the neuronal component of SBMA mice were obtained. Additionally, the potential involvement of HSPB2 in ALS remains obscure.

HSPB3 is the smallest HSPB, but normally exists in the form of trimers or tetramers [109,110,111,112]. HSPB3 also forms 1:3 stoichiometric complexes with HSPB2 [104,105], where HSPB3, with low chaperone-like activity, regulates HSPB2 function [103,113]. HSPB3 expression is elevated in cardiac and skeletal muscles but it is also present in smooth muscle cells and in different cell types of the central nervous system, including MNs [104,114,115]. Like HSPB2, HSPB3 does not respond to heat shock stresses, but it is highly responsive to proteotoxic stress. HSPB3 cellular localization is cytoplasmatic, and mainly associates with cytoskeleton components [114,115]. HSPB3 mutations cause neuropathies and myopathies [116,117], but very few studies have linked HSPB3 to MNDs. The most relevant study in this field demonstrated that, like HSPB2, HSPB3 transcription is enhanced at the symptomatic stage in skeletal muscle (quadriceps), but not in the spinal cord of a mouse model of SBMA [11]. Like HSPB2, the lack of variation in HSPB3 levels in MNs remains uncertain since few MNs are present in spinal cord samples, which are particularly enriched in other cell types.

### 3.3. HSPB4 and HSPB5

Limited implications in MNDs have been described for HSPB4 and HSPB5, which are mainly expressed in lenses [118,119], where they play an anti-aggregating role to protect from damaged proteins, assuring lens transparency [120,121,122,123,124,125]. In addition to their chaperone functions, like HSPB1, HSPB4 and HSPB5 exert anti-apoptotic roles, since they are able to block Bax, Bcl-Xs, and caspase-3 activities [122]. Of note, HSPB5 is also expressed in cardiac and skeletal muscles and in the brain [126,127,128] where it is mainly confined in glial cells [129,130,131,132,133,134,135]. The impact of mutations of these two HSPBs strongly reflects their pattern of expression and thus their tissue-specific functions, and it mainly correlates with the accumulation of proteinaceous material either of the affected protein or because of the reduced anti-aggregation properties of the mutant protein. In fact, HSPB4 mutations correlate with altered eye functions [136,137,138,139,140,141,142], while HSPB5 mutations may cause cataracts, myopathies, and cardiomyopathies [143,144,145]. The anti-aggregation activity of HSPB5 has also been described against some misfolded proteins causative of different NDs (e.g., alpha-synuclein (α-syn), mutated SOD1, and amyloid β (Aβ)) [146,147,148,149]. HSPB5 was found elevated in astrocytes and oligodendrocytes of tg ALS (SOD1-G93A) mice [150] and its overexpression prevented mutant SOD1 aggregation in vitro [150] and in cultured muscle cells [151]. Despite this, HSPB5 overexpression was unable to modify the onset and progression of the disease in two different tg ALS (SOD1-G93A and SOD1-L126Z) mice [152], suggesting that this chaperone is not implicated in ALS and may not represent a therapeutic target against the disease.

### 3.4. HSPB6

HSPB6 is widely expressed in different tissues with the highest levels in smooth muscles and in cardiac and skeletal muscles [153,154]. Similar to other HSPBs, HSPB6 can form homodimers and tetramers in solution [155,156,157,158], but it is also able to form complexes with HSPB1, HSPB5, and HSPB7 [159,160,161,162,163,164,165], and to bind the co-chaperone BAG3 [103,113,166,167,168,169]. Like HSPB1, HSPB4, and HSPB5, HSPB6 not only has a strong chaperone activity, but it is also implicated in the regulation of apoptosis, due to its capability to inhibit Bax translocation to mitochondria [170]. However, its main function is related to the regulation of cytoskeletal dynamics in muscle relaxation [171,172,173,174,175]. HSPB6 anti-apoptotic activity may explain its role in cardioprotection [170,176]. Notably, the mutations of HSPB6 described so far are all causative of different forms of cardiomyopathy [177,178].

As other HSPBs, HSPB6 is present in the brain during development. Oxidative stress and osmotic shock are potent inducers of HSPB6 in this area [179,180]. HSPB6 is not expressed in the spinal cord neurons [134], while it is generally elevated in activated astrocytes in affected brain regions of patients with specific forms of NDs [181,182] including ALS patients [134]. Notably, HSPB6 is not upregulated in neuronal cells of the spinal cord in ALS patients [134], while it appears to be upregulated in hippocampal neurons in rats in response to seizures mimicking epilepsy [183]. Therefore, this protein is likely to play a role in the non-cell autonomous response in NDs and its protective function possibly consists of preventing the aggregation of misfolded proteins, particularly α-syn and Aβ [106,107,184,185,186].

### 3.5. HSPB7

HSPB7 exerts a primary role in the cardiovascular system [187]. Like other HSPBs, it forms large oligomers [165] and it is capable of forming heteromeric structures with both HSPB6 and HSPB8 [165,188]. HSPB7 protein mainly localizes in heart and in skeletal muscle, even if high levels of its transcript are detectable in the adipose tissue [187]. Despite the fact that HSPB7 prevents aggregation and enhances autophagic degradation of misfolded proteins causative of NDs via autophagy [189,190,191], its role as protective agent in these disorders is unclear since it is poorly expressed in the brain even in response to stress. Thus, HSPB7 likely does not play a role in MNs in MNDs. HSPB7 expression has been found in myogenic areas of the skeletal muscle [187], where this chaperone interacts with structural components (actin, filamin C, and titin) [192,193,194,195]. Because of that, HSPB7 may have a surveillance function on the cytoskeleton, which increases with age, possibly to counteract aging-associated sarcopenia [128]. It is still unknown whether these functions of HSPB7 also play a role in MNDs.

Notably, no HSPB7 mutations have been reported so far.

### 3.6. HSPB8

Among all HSPBs, HSPB8 is certainly the most relevant in MN protection and, potentially, it is a candidate target for genetic or pharmacological MND treatment. HSPB8 is expressed in almost all tissues, with higher levels in cardiac and skeletal muscle tissues [22,196]; relatively high amounts of HSPB8 are present in the brain, including MNs [197,198]. Like some other HSPBs, HSPB8 expression is enhanced by several stressors, especially those related to proteostasis alterations. Proteasome saturation and blockage are likely the most potent triggers of HSPB8 induction. Indeed, proteasome inhibitors robustly enhance HSPB8 transcript and protein levels in a series of different cell types, including MNs [197,198,199,200,201,202,203]. Other cell stressors able to transcriptionally upregulate HSPB8 are sodium arsenite, oxidants, and osmotic stresses [180]. Of note, in MNs, HSPB8 expression is potentiated by the presence of misfolded proteins causative for ALS [197,198], possibly to protect against protein aggregation and proteotoxicity [198,199,204,205,206,207,208] (see below). For instance, HSPB8 is robustly increased in spinal cord MNs that survive at end stage disease in tg ALS (SOD1-G93A) mice [197,198], and in the spinal cord of ALS patients [209]. HSPB8 is also increased in the skeletal muscle of ALS and SBMA mice [9,11,210,211]. Several studies have demonstrated the pro-degradative activity of HSPB8 on several neuropathogenic proteins [106,186,198,199,205,212,213,214,215,216]. Even though HSPB8, similar to the other HSPBs, may form oligomers, its main feature resides in the capability to specifically interact with the co-chaperone BAG3 [167,188,217,218,219], which robustly enhances HSPB8 protein stability. BAG3 also serves as scaffold for the interaction with HSP70 bound to the E3-ubiquitin ligase CHIP/STUB1 [220,221]. The resulting heteromeric complex drives a selective autophagy pathway named “chaperone-assisted selective autophagy” (CASA) [30,31,220,222,223,224,225]. This mechanism physiologically preserves skeletal muscle tissues after extensive physical exercise [220], since CASA is in charge of the clearance of oxidated/carbonylated proteins and damaged components of the Z-disc structures (e.g., filamin) [224]. Apart from their physiological activity in muscle, HSPB8 and BAG3 exert other activities, since they regulate the eIF2α kinase heme-regulated inhibitor and eIF2α [226,227] during the integrated stress response of the cytosolic unfolded protein response; this allows protein synthesis blockage if proteotoxic events occur during translation. In line with this mechanism, several data have clarified that HSPB8 plays a role in stress granule composition and dynamics [53,208]. In this context, HSPB8 localizes to stress granules maintaining their dynamic state; then, HSPB8 recruits BAG3 and HSP70 to remove defective polypeptides and proteins. Similarly, when folding/re-folding of damaged proteins cannot be achieved, the CASA complex routes polyubiquitinated misfolded substrate to autophagy receptors, such as the SQSTM1/p62 [228,229,230] for LC3-II-mediated insertion into autophagosomes. The intracellular localization of CASA clients is extremely variable (especially in MNs characterized by long axons), but their delivery to autophagosomes is assisted by an active retrograde transport mediated by the motor protein dynein [199,231], which routes the CASA complex and cargo along microtubules to the microtubule organizing centre [199,230,232,233,234]. It is believed that most (but not all) autophagosomes are assembled in this cell region, where aggresomes also locate [235,236,237,238,239,240]. This pathway is an arm of the so-called routing system, that maintains the equilibrium between the autophagic and the proteasome degradation of misfolded proteins. In case of proteasome overwhelming or impairment, misfolded proteins are routed to autophagy and vice versa [241,242]. The alternative routing of misfolded proteins to proteasome requires BAG1 (a NEF of the same family of BAG3), which also binds the HSP70/CHIP complex [199,229,230,243]. While proteasome inhibition results in a robust induction of HSPB8 and BAG3 [198], autophagy blockage results in enhanced BAG1 expression [199], thus, molecularly regulating the equilibrium between the two degradative pathways. A main limitation of this system (BAG1/HSP70/CHIP for the proteasome and HSPB8/BAG3-HSP70/CHIP for CASA) is that the proteasome only degrades monomeric misfolded proteins, thus, protein aggregation heavily affects this delicate equilibrium. Of note, apart from HSPB8, all other CASA complex components are constitutively expressed at relatively high levels in most cells, while the increased expression of HSPB8 alone is sufficient to robustly enhance misfolded protein clearance via autophagy [9,198,199,204,205,206,207,211,216,223]. Indeed, when overexpressed in ALS and SBMA cell models, HSPB8 restores an impaired autophagy flux to normal levels and it enhances the degradation of disease-associated misfolded protein and aggregates of mutant SOD1, two disease-associated fragments of TDP-43 (TDP-35 and TDP-25), the RAN-translated dipeptide repeats of the C9ORF72 G_4_C_2_ expansion, and ARpolyQ. Conversely, HSPB8 down-regulation results in a dramatic accumulation of the same misfolded protein substrates in MN cells [197,199,205,206,207,216]. Thus, HSPB8 should be considered the “limiting factor” for complex assembly and function and its induction may have therapeutic relevance. *Drosophila melanogaster* models based on modulation of the functional ortholog of *HSPB8* (*HSP67Bc*) in flies expressing various forms of the TDP-43 [244] have indeed proved the therapeutic potential of HSPB8 induction in living animals [205]. A *HSPB8* knockout mouse was also recently developed, but surprisingly, no major alterations of mice survival and motor behaviour were observed [245], possibly because in physiological conditions, the absence of HSPB8 could be compensated by alternative mechanisms. Another transgenic mouse developed to overexpress an HSPB8 mutation related to CMT disease (see below) was indeed characterized by protein accumulation and marked alteration of the motor behaviour associated with the degeneration of peripheral nerves and muscle atrophy [245]. Thus, these HSPB8 mutations are responsible for a GOF of this chaperone, explaining their role in some MNDs or neuromuscular diseases, such as dHMNs, CMT2, and myopathies [168,245,246,247,248,249,250,251,252,253,254,255,256,257,258,259,260,261,262,263,264].

It is therefore crucial to find compounds that modulate HSPB8 levels in affected cells to be differentially used in MNDs (e.g., ALS and SBMA) or in mutant HSPB8-associated disorders. Interestingly, estrogens act as enhancers of HSPB8 expression [265,266], possibly explaining why gender differences characterize several NDs, including ALS [24,267]. In a large screening aimed at identifying commercially available drugs able to enhance HSPB8 expression in MNs [206], we found colchicine and doxorubicin as capable of counteracting ALS-associated misfolded protein accumulation in a HSPB8-dependent manner. Doxorubicin is relatively cytotoxic and may be employed in cancer; colchicine is well-tolerated and also regulates other genes involved in autophagy activation (transcription factor EB (TFEB), SQSTM1/p62, LC3), supporting that colchicine could represent a useful candidate to be tested in other misfolded protein-associated NDs [206,268]. Colchicine is now in a phase II clinical trial for ALS [268]. We also found that trehalose, an autophagy activator that triggers the lysosomal-mediated TFEB pathway [269,270,271], and its derivatives lactulose and melibiose [270,272,273] enhance HSPB8 (and BAG3) expression [270]. Notably, trehalose counteracts disease progression in numerous mouse models of NDs [274,275,276,277,278,279,280,281,282,283,284,285,286,287]. Other HSPB8 inducers are those capable of activating HSF1 (the regulator of HSP family gene expression), but unfortunately their selectivity toward HSPB8 is limited, as for the geranylgeranylacetone (GGA) [262,288] (which also potentiates HSPB1 expression [288,289]) or the *N*-((5-(3-(1-benzylpiperidin-4-yl)propoxy)-1-methyl-1H-indol-2-yl)methyl)-*N*-methylprop-2-yn-1-amine (ASS234) [290], already tested against some NDs [291,292,293,294,295,296]. Of note, it cannot be excluded that an enhanced expression of more than one HSPB might potentiate the therapeutic effect. These molecules may represent good candidates in combinatorial treatment designed to improve the efficacy of gene and cell -based therapeutic approaches (see below). 

Finally, HSPB8 and BAG3 together are involved in the control of the cell cycle during mitosis and cytokinesis, where they maintain proper actin structure homeostasis and dynamics [166,250,297,298]. HSPB8 is also involved in RNA metabolism, thanks to its ability to bind DDX20, an RNA helicase involved in the activity of the SMN protein; notably, some HSPB8 mutants show alteration of DDX20 binding [299].

### 3.7. HSPB9 and HSPB10

Finally, the last two members of the HSPB family are HSPB9 and HSPB10. No data are available on their involvement in MNDs or NDs in general, but, at least for HSPB9, a chaperone activity against ND-related substrates has been confirmed in vitro [189]. Nevertheless, both HSPBs are expressed almost exclusively in testis [30,31,63,300,301], and, so far, there are no reports showing that they may be induced in response to proteotoxic (or other) stresses in the brain or muscle.

## 4. Conclusions

In conclusion, the HSPB family has been extensively studied in the past two decades and the overall role of its members dissected out and characterized. These activities span from a common and distinctive function of HSPBs in the refolding and anti-aggregation processes to the activation of the degradative pathways that are limited to some members of the family. Despite these common activities against well-known substrates linked to NDs (Figure 3), only some members of the HSPB family have been clearly implicated in NDs and specifically in MNDs. Among these, the best candidates to be deeply studied to tackle MNDs are HSPB1 and HSPB8. The contribution of other HSPBs (e.g., 2,3,4,5) has to be fully unravelled, since some are poorly expressed in the brain, but highly expressed in muscle, which is presently considered a potentially relevant player in the pathogenesis of some MNDs. Of note, small molecules capable of inducing HSPB8 expression have already been reported and are now under clinical evaluation. Less is known for HSPB1. These potential therapeutic approaches may also be useful in the design of novel combinatorial treatments that are emerging in recent years to treat misfolded protein-associated MNDs. As few examples, other more specific strategies, such as gene and cell therapies are promising approaches for these diseases [302,303,304]. These approaches rely on the use of stem cells and of vectors conceived on the bases of the recently approved gene therapies adopted for SMA [305], as well as on the use of synthetic antisense oligonucleotides specifically targeting the mRNA transcript encoding the mutant protein responsible for a given MND [306,307,308,309,310]. The present limitation of their use is the capability to deliver the various vectors or nucleotides in the affected tissues in order to be taken up from cells involved in the disease. In this view, the classical drug-based therapies, although less specific, generally assure good pharmacokinetic results; therefore, they may likely assist and improve the gene-based approaches.

However, despite the many findings in recent years that have contributed to understanding the mechanism of action of HSPB8 and HSPB1 in response to proteotoxic events, some aspects of their regulation and function, as well as its implication in human cancer, remain partially unknown. Therefore, there is the need to better understand how these two HSPBs contributes to regulating cell parameters in different cell types prior to utilizing the strategy to modulate their expression genetically or pharmacologically for therapeutic purposes in a large population of affected individuals.

## Figures and Tables

**Figure 1 ijms-23-11759-f001:**
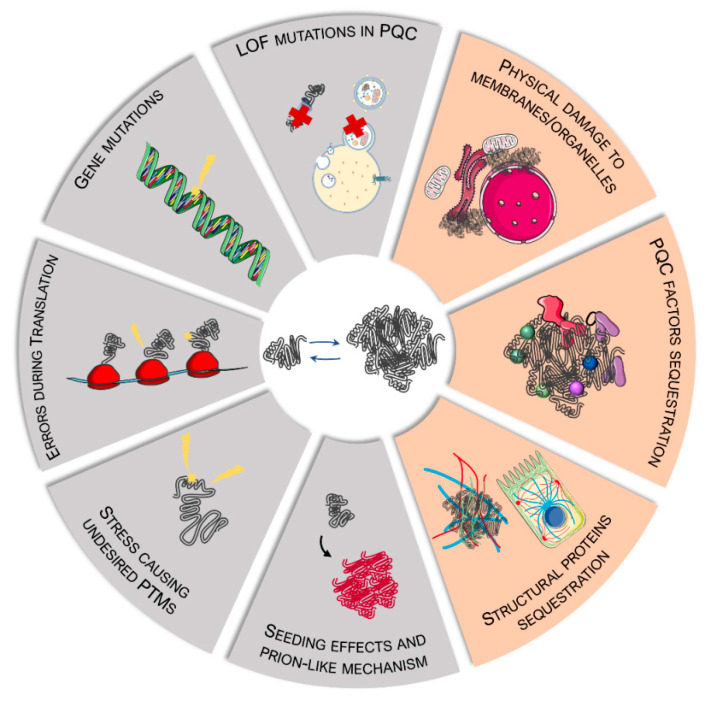
Mechanisms and consequences of protein misfolding and aggregation. In grey, mechanisms that directly or indirectly cause protein misfolding and accumulation into aggregates. In orange, the main damaging effects of misfolded proteins and aggregates on cell homeostasis. LOF = loss of function; PQC = protein quality control; PTMs = post-translational modifications. This figure was created using Servier Medical Art templates, which are licensed under a Creative Commons Attribution 3.0 Unported License; https://smart.servier.com (accessed on 12 October 2021).

**Figure 2 ijms-23-11759-f002:**
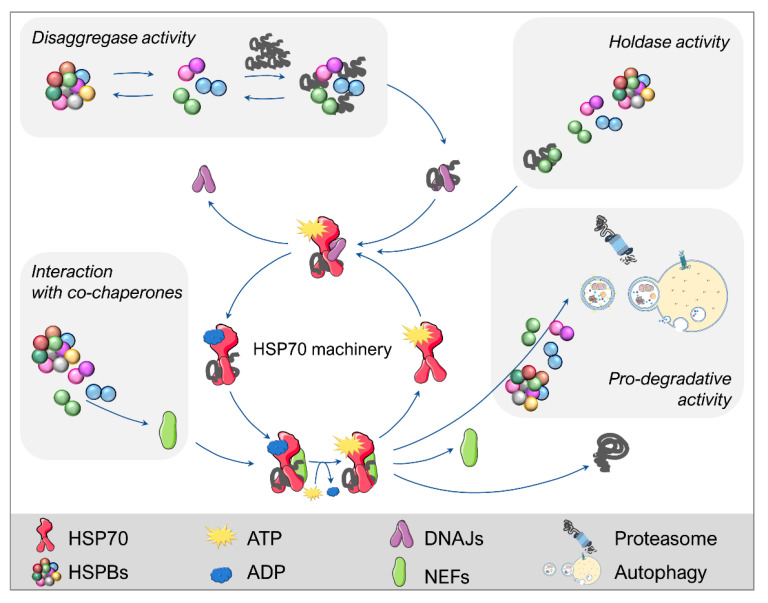
HSPBs participation in the PQC system. The HSP70 machinery is the core system of the proteostasis network. HSP70s perform folding cycles using ATP and collaborating with other chaperones (e.g., DNAJs) and co-chaperones, such as NEFs. HSPBs directly or indirectly interplay with HSP70 by exerting a disaggregase activity and by keeping client proteins in a refolding-competent state (holdase activity). Additionally, some HSPBs cooperate with co-chaperones and/or facilitate the routing of client proteins to degradation, which can occur through autophagy and the proteasome. This figure was created using Servier Medical Art templates, which are licensed under a Creative Commons Attribution 3.0 Unported License; https://smart.servier.com (accessed on 12 October 2021).

**Figure 3 ijms-23-11759-f003:**
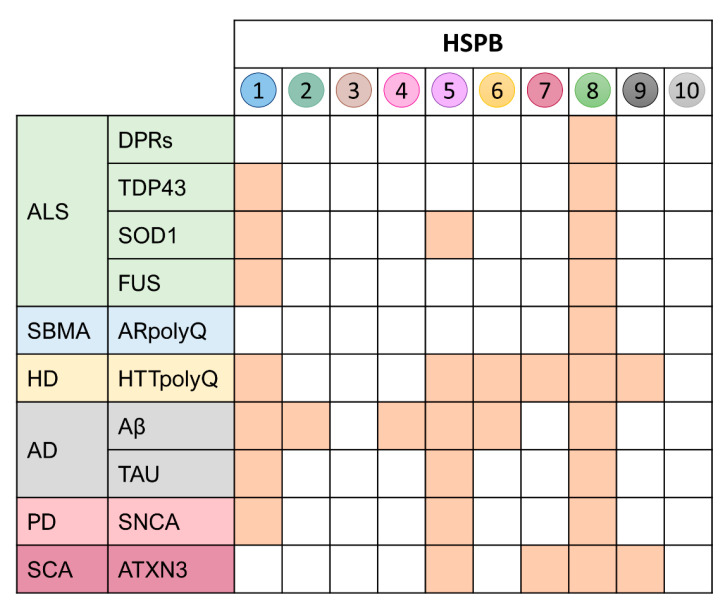
HSPBs anti-aggregating and/or pro-degradative functions against MND/ND substrates. This graph outlines the substrates against which HSPBs have been found to exert their activity (orange squares).

## Data Availability

Not applicable.

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
