# Peer review of "The Role of Small Heat Shock Proteins in Protein Misfolding Associated Motoneuron Diseases"

_ijms, 2022, doi:10.3390/ijms231911759_

Round 1

Reviewer 1 Report

In this manuscript, the authors discuss how protein misfolding and aggregation could play an important role in motoneuron disease pathology. Among this theme, the authors highlighted the HSPBs family and how they contribute to the HSP70s-driven management of misfolded proteins. The authors then discussed all ten members of the HSPB family. Among these discussions, the authors provide detailed information on expressions, functions, how or if each of them is linked to MNDs and potential molecular mechanisms in refolding and anti-aggression process. The authors are knowledgeable in literature and provided information regarding if there are mutation studies for each member of HSPBs.

Some minor suggestions:

Please restrain using abbreviations, as they break down sentences and could cause confusion to the readers. When some terms are never mentioned later, there is no need to include their abbreviation (for example Line 60 ASO). Where the abbreviations are used in the paragraph that is far from its first introduction, please include the full description (e.g., SBMA was introduced in line 33 and then used in line 115).

Might be worth discussing in the conclusion sections how other members besides HSPB8 could be working as potential therapeutic targets for neurodegeneration diseases or MNDs.

Overall, the manuscript is a valuable source of knowledge and I suggest accepting this manuscript for publishing.

Author Response

We are grateful to Reviewer #. 1 for the very positive comments he/she provided on our review. We regards to the minor suggestion raised we are pleased to answer as follow:

Q1 - Please restrain using abbreviations, as they break down sentences and could cause confusion to the readers. When some terms are never mentioned later, there is no need to include their abbreviation (for example Line 60 ASO). Where the abbreviations are used in the paragraph that is far from its first introduction, please include the full description (e.g., SBMA was introduced in line 33 and then used in line 115). 

Thanks for the suggestion. We deleted all unnecessary abbreviations, as suggested by the Reviewer.

Q2 - Might be worth discussing in the conclusion sections how other members besides HSPB8 could be working as potential therapeutic targets for neurodegeneration diseases or MNDs. 

Sorry we missed this aspect in our conclusion. We now better discussed how some other HSPBs could represent a therapeutic target in MNDs.

Reviewer 2 Report

The review article by Tedesco et. al. elaborates the role of protein quality control failure in motor neuron diseases, with particular focus on a relatively understudied class of chaperones, the small heat shock proteins. It is well organized and covers quite a bit of ground in this topic and it should be a valuable resource as a review of primary literature. I have a few comments and suggestions for improving the current manuscript, so it can be even more valuable to readers.

1.     The review would greatly benefit from including a discussion and reference to a transcriptional profiling type study of motor neurons, particularly of MND models.

2.     The introduction summarizes the clinical features of Motor Neuron Diseases, but doesn’t mention Parkinsonism, which is often a common feature of MNDs.

3.     The interplay of chaperone network with autophagy/lysosome-mediated degradation has been mentioned in the review. However, the specific features of substrates, their cellular locations, PTMs or the role of various cofactors in diversion of misfolded cargo to one system or the other, could be elaborated better, as it has been done in case of CASA.

4.     The authors have mentioned the use of doxorubicin and colchicine as current therapeutic directions, under evaluation for enhancing HSPB8 activity. While there have been some initial promising results, these are quite non-specific therapeutics and this caveat ought to be mentioned along with other strategies such as gene therapy and enzyme replacement.

5.     The quality of the figures especially Figure 2, which is a key figure, requires a more details. In the current format it is understandable but a bit too minimalist and lacking sufficient detail to be a snapshot of a large and complex field.

6.     The review is overall well-written, but the abstract could benefit from further editing and corrections.

Author Response

We thank Reviewer # 2 for the very positive comments he/she provided on our review and for the suggestion provided to improve the overall quality of the manuscript.

We are pleased to say that we were able to reply to all questions raised as follow:

Q1 - The review would greatly benefit from including a discussion and reference to a transcriptional profiling type study of motor neurons, particularly of MND models.

Thanks for the suggestion. We added two paragraphs in which we discussed (and provided references) transcriptional profiling type studies that have been performed in the field of MNDs. The number of studies, the different type of MNDs tissue/models used for the analysis, the different time window in which these comparisons were done, make very difficult to extrapolate data that can be provided in a review dedicated to the role of HSPBs. However, we considered useful to mention some review studies in which these comparisons were done among various cell models for study the different MNDs. Thus, readers will be able to find suitable information on the transcriptional changes occurring in MNDs and models. 

2Q.     The introduction summarizes the clinical features of Motor Neuron Diseases, but doesn’t mention Parkinsonism, which is often a common feature of MNDs.

The comment is correct. We now added a sentence in which we have mentioned Parkinsonism as one of the feature characterizing some MNDs.

Q3.     The interplay of chaperone network with autophagy/lysosome-mediated degradation has been mentioned in the review. However, the specific features of substrates, their cellular locations, PTMs or the role of various cofactors in diversion of misfolded cargo to one system or the other, could be elaborated better, as it has been done in case of CASA.

We thank the reviewer for the suggestion. As we discussed CASA in the context of HSPB8 role, we added a few sentences on how HSPB1 responds to stress, its intracellular relocation and interplay with other cofactors. Additionally, we discussed its activity in substrates that are involved in liquid-liquid phase separation.

Q4.     The authors have mentioned the use of doxorubicin and colchicine as current therapeutic directions, under evaluation for enhancing HSPB8 activity. While there have been some initial promising results, these are quite non-specific therapeutics and this caveat ought to be mentioned along with other strategies such as gene therapy and enzyme replacement.

Thanks for this suggestion. We mentioned the other strategies presently under investigation or in use in our revised manuscript. The paragraph was added in the conclusion section in order to provide a wider view on the potential therapeutic approaches for protein misfolding associated MNDs.

Q5.     The quality of the figures especially Figure 2, which is a key figure, requires a more details. In the current format it is understandable but a bit too minimalist and lacking sufficient detail to be a snapshot of a large and complex field.

Thanks for the suggestion. We added more details to figure 2 as requested and we hope now better provide a view of the complexity of this field, with particular focus on HSPBs activities in proteostasis. With regards to the quality of the other figures, we believe the problem is due to the conversion from .doc to .pdf files from the MDPI web system. Better quality figures have been uploaded in the submission site.

Q6.     The review is overall well-written, but the abstract could benefit from further editing and corrections.

Thanks for noting this. We extensively edited the abstract as suggested.